# The Use of Hydroxyapatite Loaded with Doxycycline (HADOX) in Dentoalveolar Surgery as a Risk-Reduction Therapeutic Protocol in Subjects Treated with Different Bisphosphonate Dosages

**DOI:** 10.3390/medicina59010046

**Published:** 2022-12-27

**Authors:** Roberto Sacco, Suelen Cristina Sartoretto, Rodrigo Figueiredo de Brito Resende, Jose de Albuquerque Calasans-Maia, Alexandre Malta Rossi, Victor Hugo de Souza Lima, Carlos Fernando de Almeida Barros Mourão, Jose Mauro Granjeiro, Julian Yates, Monica Diuana Calasans-Maia

**Affiliations:** 1Oral Surgery Department, School of Medical Sciences, Division of Dentistry, The University of Manchester, Coupland 3 Building, Oxford Rd, Manchester M13 9PL, UK; 2Oral Surgery Department, Dental School, Fluminense Federal University, Rio de Janeiro 24020-140, Brazil; 3Orthodontic Department, Dental School, Fluminense Federal University, Rio de Janeiro 24020-140, Brazil; 4Brazilian Center for Research in Physics, Applied Physics and Nanoscience, Department of Condensed Matter, Rio de Janeiro 22290-180, Brazil; 5Graduate Program, Faculty of Sciences and Biotechnology, Fluminense Federal University, Niteroi 24210-201, Brazil; 6Department of Periodontology, Dental Research Administration, Tufts University School of Dental Medicine, Boston, MA 02111, USA; 7National Institute of Metrology, Quality and Technology (INMETRO), Duque de Caxias, Rio de Janeiro 25250-020, Brazil

**Keywords:** bisphosphonate, MRONJ, hydroxyapatite, doxycycline, drug-delivery system

## Abstract

Medication-related osteonecrosis of the jaw (MRONJ) is considered as a severe adverse side effect of specific drugs such as anti-resorptive and anti-angiogenic medications. Evidence suggests that MRONJ is linked to invasive dental procedures, mainly dentoalveolar surgery. Several preventive strategies to minimize the risk of developing MRONJ have been investigated. However, no investigation has been attempted to evaluate the therapeutic effect of local drug-delivery technology as a preventive strategy protocol. The aim of this study is to evaluate the efficacy of hydroxyapatite-containing doxycycline (HADOX) in rats with high-risk MRONJ development. All the rats used in this study were divided into seven groups. Six groups of rats out of seven were exposed to two different doses of antiresorptive drug therapy for four weeks before undergoing an upper incisor extraction. After 28 days, all the animals were euthanized, and the bone blocks were processed for histological and histomorphometrical evaluation. The histomorphometric analysis confirmed that newly formed bone (NFB) was present in all groups, with significant differences. NFB in the HADOX group treated with zoledronic acid at 4% showed (28.38; C.I. 22.29–34.48), which represents a significant increase compared to HA (15.69; C.I. 4.89–26.48) (*p* = 0.02). A similar pattern was observed in the HADOX group treated with zoledronic acid 8% ZA treatment (*p* = 0.001). *Conclusions*: HADOX did not inhibit any bone repair and reduced early inflammatory response. Hence, HADOX could promote bone healing in patients undergoing antiresorptive drug therapy.

## 1. Introduction

Antiresorptive drugs such as bisphosphonate (BP) or denosumab (DMAB) are largely used to treat a range of benign and malignant skeletal disorders, including multiple myeloma, bone metastatic disease, and osteoporosis [1,2].

BPs and DMAB reduce the bone resorption via different pathways. BPs inhibit the resorption of bone by osteoclasts and may also have an effect on osteoblasts. Structurally, BPs are similar to pyrophosphate and have a very high affinity for bone mineral because they bind to hydroxyapatite crystals. More recently, it has been advised that BPs also function to limit both osteoblast and osteocyte apoptosis [3,4]. The relative importance of this function for bisphosphonate activity is currently unclear. However, it is known that BPs bind to and inhibit the activity of farnesyl pyrophosphate synthase, which in turn leads to osteoclast apoptosis by generating the cytotoxic analogues of adenosine triphosphate (ATP), which interfere with the mitochondrial function of osteoclasts [5].

The DMAB mechanism of action is based on the complex interaction of the receptor activator of nuclear factor kappa B (RANK), RANK ligand (RANKL), and osteoprotegerin (OPG). DMAB is a human monoclonal antibody to RANKL that acts in a manner similar to OPG by blocking RANKL and inhibiting osteoclastogenesis [6].

These medications are associated to a serious condition called MRONJ, which was first reported in 2003 [1,7]. MRONJ often presents as a region of necrotic bone exposed through the oral mucosa in the jaw [8,9].

MRONJ is reported to develop in approximately 8% of the oncology patients taking high-dose BP or DMAB and in 0.01% of osteoporotic patients taking low-dose BP or DMAB [10,11].

The management of MRONJ can be extremely challenging, and the different therapeutic approaches recommended in the literature are often contradictory and not strongly supported by solid evidence [12,13]. Hence, preventive strategies have been suggested to mitigate the onset of MRONJ. These strategies include dental screening prior to commencing the drug therapy, prophylactic dental extraction, drug holiday, antibiotic prophylaxis, and a number of intra-operative surgical procedures [1,14,15].

Intra-operative risk-reduction strategies are particularly important to assess as currently there remains little guidance for clinicians with respect to performing dental extraction in individuals at risk of MRONJ. Of note, dental extraction still represents a fundamental treatment for ongoing and otherwise not curable dental infections, which carry a high-risk factor for the development of MRONJ [16]. A recent systematic review has suggested that the risk of developing MRONJ in cancer individuals exposed to high-dose BPs and undergoing dental extraction may be reduced through an adjusted extraction protocol [17].

In this regard, it has never been considered to use a synthetic biomaterial applied on its own or as a carrier for the drug-delivery system, which could be used as a preventive strategy in patients at high risk of MRONJ that require dental extraction. Indeed, regenerative medicine has tried to develop biomaterials with the purpose of promoting tissue reconstruction and providing better healing [18]. Hydroxyapatite (HA) is a bioactive and biocompatible material that has been widely used in oral and maxillofacial surgery. HA mineral matrix, which resembles the crystallographic structure of natural bone, is commonly used as a synthetic bone-graft material [19]. Indeed, HA has an excellent affinity for biological substances such as proteins, enzymes, and cells and has been previously used as a delivery-system drug due to its non-toxicity and excellent biocompatibility [18,20,21].

The controlled delivery of therapeutic substances by hydroxyapatite has been proposed as a viable approach to prevent and control inflammatory processes and chronic infections and to enhance the endogenous healing capacity of bone defects [22,23,24].

The physical and chemical properties of this material (such as porosity, interconnected pores, and high surface and volume ratio) leading to its effectiveness as a carrier for delivery of therapeutic agents (such as drugs, genes, antigens, enzymes, and other proteins) has been previously described. [25,26].

The use of doxycycline (DOX) has previously been positively documented in clinical studies for its broad-spectrum therapeutic effect [27]. In addition to its antibacterial action, by inhibiting microbial protein synthesis studies have also shown beneficial effects on bone regeneration. These effects are mainly explained by its inhibitory properties on inflammation and osteoclastogenesis [27,28].

Among the secondary properties presented by DOX, the anti-inflammatory activity has already been implemented in the treatment of diseases related to chronic inflammation. The FDA recommends its use in the treatment of rosacea and acne vulgaris [29]. The ability to reduce inflammation has been shown to be beneficial for bone regeneration since inflammation tends to stimulate bone resorption by activating osteoclasts. Doses of 10 and 25 mg.kg^−1^ were shown to be capable of inhibiting inflammatory infiltration, and increased bone formation was observed in treated animals [30].

The objective of this study is to investigate the in vivo alveolar bone repair and the ability of hydroxyapatite-containing doxycycline (HADOX) to reduce the risk of MRONJ development after tooth extraction in a group of rats exposed to different BP dosages.

Our hypothesis was that the physical differences between HA and HADOX might have led to inflammatory and regenerative proliferation in vivo. Hence, HA and HADOX would have improved the healing time process equally in both the subjects exposed to different doses of antiresorptive drugs.

## 2. Materials and Methods

### 2.1. Ethics Considerations

The animal breeding and experiments were performed according to the conventional guidelines of the NIH Guide for the Care and Use of Laboratory Animals and following the Brazilian Directive for the Care and Use of Animals for Scientific and Didactic Purposes—DBCA and the CONCEA Euthanasia Practice Guidelines. The research protocol of this work was approved by the Ethics Committee of Animal Use from Fluminense Federal University (CEUA/UFF) under protocol number 2816310120. This study follows the ARRIVE guidelines supplemented by PREPARE regarding relevant items [31,32].

### 2.2. Animal Characterization and Preparation

The study was conducted on 35 female Wistar rats (*Rattus norvegicus*, *albinus*) weighing approximately 200–250 g. These animals were kept in the animal experimentation laboratory at Universidade Federal Fluminense, School of Dentistry, and maintained on a 12:12 h light/dark cycle (lights on at 7:00 AM) at 22 ± 2 °C with ad libitum access to a standard laboratory diet and water.

A pilot study with 3 animals was performed prior to conducting the full study and reported no adverse effects or experimental issues. The animals used in the pilot study were eventually included in the final animal study’s number as per recommendations of the NC3Rs Program (https://nc3rs.org.uk/3rs-resources/conducting-pilot-study; accessed on 10 November 2022).

The animals were kept in mini-isolators (*n* = 5) identified according to the group and study periods; the mini-isolators were cleaned daily. The animals received a grained solid food before the study (except for 12 h before and after surgery) and water *ad libitum*. The Wistar rats were randomly distributed into a total of seven groups through online random number generators: G1, Zoledronic acid 0.04mg and HA without doxycycline, (HA 4% ZA); G2, Zoledronic acid 0.08mg and HA without doxycycline (HA 8% ZA); G3, Zoledronic acid 0.04mg and HA with doxycycline (HADOX 4% ZA); G4, Zoledronic acid 0.08mg/kg and HA with doxycycline, (HADOX 8% ZA); G5, Zoledronic acid 0.04/mg without graft (Sham 4%); G6, Zoledronic acid 0.08/mg without graft (Sham 8%) and G7, clot without zoledronic acid treatment (Sham). Zoledronic acid was injected subcutaneously once a week for four weeks. One week after the last administration, each group received an extraction of an upper incisor. In the zoledronic acid groups with local antibiotics (G3 and G4), the sockets were filled with doxycycline-containing hydroxyapatite microspheres (HADOX). In the control group, the incisors were extracted, and the sockets were filled with HA without doxycycline (G1 and G2) and SHAM without graft biomaterial (G5, G6 and G7). All the animals were euthanized after four weeks following the teeth extractions, as described below, for a histological and histomorphometric analysis to establish the entity of the osteonecrosis and for alveolar bone healing, see Figure 1.

### 2.3. Sample Size Calculation

The sample size was calculated based on a pilot study considering 90% for the power of analysis and a significance level of 5%, with 5 animals/group being suggested (https://www.sealedenvelope.com/power/continuous-superiority/; accessed on 1 August 2022). The randomization of animals was performed on the website (https://www.sealedenvelope.com/simple-randomiser/v1/lists; accessed on 1 August 2022).

### 2.4. Bone-Substitute Synthesis and Characterization

HA powders were produced via the wet method by dropwise addition of Diammonium hydrogen phosphate solution to a calcium nitrate solution at 37 °C. The pH was adjusted to 12 and dried at 50 °C in a drying oven. An X-ray Fluorescence Spectrometer (PW 2400, Philips Analytical X-ray) was used to determine the Ca/P ratio. The physicochemical characterization was conducted by performing Fourier-transform infrared spectroscopy (FTIR) (Shimadzu IRPrestige-21) and X-ray diffraction XRD (PANalytical X’Pert). HA powder had a Ca/P ratio of 1.66 and XRD peaks typical of crystalline hydroxyapatite without the contribution of contaminant phases (Figure 2A). The FTIR spectrum showed OH- (3570 cm^−1^ and 630 cm^−1^) and (PO4)3- (1090 cm^−1^, 1054 cm^−1^ 962 cm^−1^ 600 cm^−1^ and 569 cm^−1^) bands confirming the HA attribution. Bands of carbonate ions impurities substituting phosphate ions on the HA surface were also detected in the region around 1450 cm^−1^ (Figure 2B).

Microsphere preparation followed the method described previously [33,34]. Briefly, HA dry powder was mixed with a 2% sodium alginate solution in distilled water. The HA/alginate solution was extruded in a 0.3 M calcium chloride solution via a 0.45 μm needle to form the Alginate/HA microspheres. After being dried, microspheres with diameters between 600 μm and 424μm were heat-treated at 1000 °C for four hours and sterilized in an autoclave. HA microspheres topography was analyzed by electron microscope JEOL JSM-6490LV (Figure 3).

The DOX adsorption assay was performed in a controlled atmosphere of a laminar flow. Previously autoclaved microspheres were immersed for 1 h in a filtered (0.22 µm syringe filter) Dox solution of 1.5mg/mL in PBS 7.4 under constant agitation. After immersion, the microspheres were separated from the DOX solution and dried using sterile 0.44 µm paper filters. The remaining solution was used to quantify the amount of DOX adsorbed in microspheres. 50 mg of microspheres containing 1.41 mg of DOX were implanted in each animal.

### 2.5. Anaesthesia and Surgical Procedures

After anaesthesia with intramuscular injection of 100 mg/kg ketamine hydrochloride (Virbac^®^, Veltbrands, São Paulo, Brazil) and sedation with 10 mg/kg xylazine (FortDodge^®^, Rio de Janeiro, RJ, Brazil), the antisepsis of the perioral region and oral mucosa was achieved with 0.12% chlorhexidine. In the absence of pain reflexes, the animal was positioned in a dorsal decubitus position on the operation table and the sterile surgical field was placed, isolating the area to be operated. Firstly, an intrasulcular incision was made in the gingival sulcus of the upper incisor with a surgical blade number 15 c (Solidor^®^, Lamedid—Osasco, São Paulo, SP, Brazil). Next, the upper incisor was dislocated with periotome (Duflex^®^, São Paulo, SP, Brazil) and extracted for implantation of biomaterials. After the surgical manipulation, the mucosa was repositioned and sutured with Nylon 5.0 thread (Mononylon^®^, Ethicon—Vila Olímpia, São Paulo, SP, Brazil). Once the surgical procedure was completed, postoperative analgesia with 1 mg/kg meloxicam (Duprat^®^, Rio de Janeiro, Brazil) was administered subcutaneously every 24 h for three days along with 10 mg/kg tramadol hydrochloride (Tramal ^®^, Pfizer, São Paulo, SP, Brazil) via SC every 12 h, also for three days.

### 2.6. Euthanasia and Histological Processing

After four weeks from surgery, the rats were euthanized (*n* = 5 per group) via a lethal dose of anaesthetic (overdose, 200 mg/kg of ketamine) (Francotar^®^, Virbac, Jurubatuba, São Paulo, SP, Brazil) and 20 mg/kg of xylazine (Sedazine^®^, Fort Dodge, Rio de Janeiro, RJ, Brazil), intramuscularly. The bone block involving the socket area was removed with a #6 long carbide spherical drill (Spherical download #6 FG^®^, FG, São Paulo, SP, Brazil), coupled to a micromotor (Micromotor Marathon 3 Champion—Talma, São Paulo, SP, Brazil), with a safety margin of 2 mm. The specimens were fixed in 4% formaldehyde, decalcified in decalcification solution (Allkimia^®^, Campinas, São Paulo, Brazil) for 48 h, and cut in the axial direction of the alveolar middle third. Afterwards, the samples were embedded in paraffin, cut at 5-μm thickness, and stained with Masson’s Trichrome (TM).

### 2.7. Descriptive Microscopic Analysis

The slides were observed in a brightfield light microscope (OLYMPUS BX43, Tokyo, Japan), and the photomicrographs were captured by a high-resolution digital camera (OLYMPUS SC100, Tokyo, Japan) using 4x achroplan objective lenses for wide viewing of the area of interest and 40× for obtaining tissue details. The software used for high-resolution capture was CELLSENS^®^1.9 (Digital Image, Tokyo, Japan) at the Clinical Research Laboratory in Dentistry (LPCO) at UFF.

The biological response to HA and HADOX was evaluated for newly formed bone, the presence of remaining biomaterial, connective tissue, and inflammatory response.

### 2.8. Histomorphometric Analysis

Histomorphometric analysis allowed the quantification of the connective tissue and the newly formed bone in the socket. The morphometric measurements were performed using the Image-Pro Plus^®^ software, version 4.5.0.29 (Media Cybernetics, Silver Spring, EUA). In each histological slice, six non-superimposing microscopic fields obtained by scanning at 20× magnification were captured in the medium third region of the socket. A grid of 250 points superimposed on the area under analysis allowed the determination of the volume density of the newly formed bone and of the connective tissue.

The density of newly formed bone was determined by the amount of the bone formed in the defect area available for bone growth (NFB/A-available), discounting the area occupied by the biomaterial (A-available = A-defect – A-biomat). Therefore, the bone density will be given by: NFB(A-available)% = NFB(A-defect)/[1-A-biomat/A-defect]. The same procedure was applied to determine the connective tissue formation rate.

### 2.9. Statistical Analysis

The Shapiro–Wilk normality test evidenced the normal distribution of the sample. The quantitative description for the newly formed bone, biomaterial, and connective tissue variables was performed through parametric analysis with means and confidence intervals at a 5% significance level. Analysis of variance (ANOVA) and Tukey’s post-test were applied to investigate the statistical differences between Sham, Sham 4%, and Sham 8% for newly formed bone. Student’s *t*-test was used to evaluate the differences between acid zoledronic 4% versus 8% with the same grafts. Moreover, this test was used to evaluate the differences between HA and HADOX variables at the same percentage of acid zoledronic acid. The analysis was performed using Graph Pad PRISM 8.3 software.

## 3. Results

### 3.1. Clinical

All animals remained in good health during the healing period. Moreover, all the rats tolerated the experiment well with no signs of extraoral bone exposure, fistula, or skin necrosis in any of the groups included in the study.

Four weeks after the surgical extraction, all rats of the HADOX groups (G3, G4) and the clot groups (G5, G6 and G7) showed good epithelial socket coverage with no signs of inflammation and/or intraoral bone exposure. In contrast, all the animals in the HA groups (G1, G2) presented some epithelial socket coverage with signs of erythema/inflammation and exudate or pus within the area of surgical site.

### 3.2. Histological Descriptive Analysis

#### 3.2.1. Sham Groups

After four weeks of surgical procedures, the Sham group (without ZA treatment) showed the defect area totally filled with a large amount of mature trabecular bone interspersed with connective tissue (Figure 4A,B). In the Sham 4% ZA (Figure 4C,D) and 8% ZA groups (Figure 4E,F), the defects were partially filled by trabeculae of newly formed bone interspersed with connective tissue. In the center of the defect, there was a predominance of connective tissue. In the Sham 8% ZA, a greater central area occupied by connective tissue was observed.

When comparing the results of the Sham group without ZA treatment and the Sham group treated with 8% ZA, a significant difference between the amount of newly formed bone and connective tissue was visible. According to this finding, the MRONJ rat model used promoted a more considerable amount of connective tissue in the socket, evidencing a delay in healing when compared to the Sham group without ZA treatment (Figure 4E,F vs. Figure 4A,B).

#### 3.2.2. Animals Treated with 4% ZA and Bone Grafts

The HA 4% ZA group presented the defect area almost filled by connective tissue, scarce biomaterial with giant cells in the periphery. It was possible to note an area of intense inflammatory infiltrate and micro-abscesses in the center of the defect (Figure 5A,B).

In the HADOX 4% ZA group, the area of interest was filled by spheres of biomaterial surrounded by connective tissue with sparse multinucleated giant cells. Also, newly formed bone trabeculae contiguous to the biomaterial was observed, presenting a centipede movement (Figure 5C,D).

#### 3.2.3. Animals Treated with 8% ZA and Bone Grafts

The HA 8% ZA group appointed rare fragments of biomaterial in the center of the defect. Delicate bone trabeculae were noted in the periphery, while the center of defect was filled by connective tissue (Figure 6A,B).

The HADOX 8% ZA (Figure 6C,D). presented a pattern-like 4% ZA. The spheres of biomaterial were distributed over the entire length of the defect. The trabeculae of newly formed bone presented an aspect of mature bone contiguous to the biomaterial.

### 3.3. Histomorphometric and Statistical Evaluations

#### 3.3.1. Newly Formed Bone

The density area of NFB in the clot groups are shown in Figure 7. The Sham group (without ZA treatment) presented significant (*p* = 0.0007) higher values of NFB values (71.51; C.I. 69.28–73.73) than the Sham 8% ZA (55.94; C.I. 46.10–65.78) four weeks after surgery. Also, the Sham 4% ZA (64.40; C.I 61.75–67.06) showed more NFB area than Sham 8% ZA (*p* = 0.0426).

The density area of NFB for the experimental groups (HA and HADOX) in rats treated with 4% and 8% of zoledronic acid, after four weeks of grafting, revealed the positive effect of doxycicline, as shown in Figure 8. HADOX 4% ZA presented higher density (*p* = 0.02) of NFB (28.38; C.I. 22.29–34.48) compared to HA 4% ZA (15.69; C.I. 4.89–26.48). The 8% ZA treatment showed the same pattern (*p* = 0.001). There were no statistical differences between 4% and 8% ZA treatment in the HA and HADOX groups (*p* > 0.05) Table 1.

#### 3.3.2. Biomaterial

At 4% ZA treatment, the HADOX showed substantially (*p* < 0.0001) more biomaterial volume (42.83; C.I. 35.21–50.54) than the HA group (5.26; C.I. 1.80–8.71). This difference also occurred at 8% ZA group with HADOX (42.06; C.I. 36.23–47.88) presenting almost 5-fold more volume of graft (*p* < 0.0001) than the HA group (3.50; C.I. 1.54–5.45). There were no significant statistical differences (*p* > 0.05) between 4% and 8% ZA treatment in the HA and HADOX remaining biomaterial (Figure 9 and Table 1).

#### 3.3.3. Connective tissue

The 4% ZA treatment presented greater (*p* < 0.0001) connective tissue volume in the HA group (79.06; C.I. 69.20–88.91) compared with HADOX (28.79; C.I. 24.20–33.38), similar to the 8% ZA treatment (*p* < 0.0001). There were no statistical differences (*p* > 0.05) between 4% and 8% ZA treatment in the HA and HADOX groups regarding the connective tissue (Figure 10 and Table 1).

## 4. Discussion

Many studies have highlighted that the ONJ is a severe condition which is very challenging to manage, especially in individuals with oncologic disease [1,8]. Further studies have reported that the major MRONJ triggering event is represented by dental extraction [35,36]. Although numerous studies have been published, there is still a lack of consensus regarding the MRONJ risk-reduction strategies [1,8,35].

This represents a major unmet healthcare need as anti-resorptive medications are widely used by individuals with metastatic cancer and osteoporosis, and a notable proportion of these individuals regularly undergo dental extraction due to concomitant periodontal disease or non-restorable dental caries or fractures [8,37].

In 2015, a systematic review evaluated as the primary objective the incidence of MRONJ after tooth extractions in patients treated or previously treated with antiresorptive drugs for different medical reasons. A secondary objective compared the results of various extraction protocols used as an MRONJ-prevention strategy; however, this review was unable to identify a valid protocol that could reduce the incidence of MRONJ [16]. Later, another systematic review reported that there was a lack of robust evidence to support any specific dentoalveolar surgery protocols able to reduce MRONJ incidence [17]. However, none of these systematic reviews investigated using graft materials as a drug-delivery material as a preventive strategy protocol to MRONJ. A preliminary search in the literature showed no previous investigation into antibiotic drug-delivery biomaterials in different drugs-exposure groups having ever been performed as an MRONJ-preventive strategy protocol.

A study by Diniz-Freitas and Limeres reported that peri-operative systemic antibiotic prophylaxis in cases of oral surgery potentially decreased the risk of osteonecrosis. However, the outcome of this study was based on low-level evidence such as case series or cohort studies [38]. On the other hand, the association of antibiotics with hydroxyapatite has been studied for the treatment of trauma and/or degenerative bone-related infectious diseases [33].

The histological findings of this study have clearly confirmed NFB around the microspheres of HA in all the groups, with a better response in the groups where HADOX was used.

The benefit of antibiotic-loaded HA on bony defects is both antimicrobial and reparative. The nanostructured HA allows the loading of antibiotics and an adequate quick biomaterial resorption. Moreover, other advantages are demonstrated by a system carried via HA which protects the drug by preventing its diffusion and degradation from body fluids and/or its retention time in the body. In this way, the loaded drug is used and concentrated in the local site. [22,39].

Potentially, the use of HADOX could be meaningful for hard-tissue procedures, such as bone regeneration and treatment of local bone disease, and as surgical bone-preventive strategies when there is insufficient bone vascularization [22]. However, the main challenges of antibiotic-loaded HA are represented by the control of antibiotics through in vivo release, the length of bactericidal action in bony defects, and toxicity [22,40].

Currently, there are many in vivo models used for MRONJ studies, with different BP dosages and durations. Unfortunately, at the present time, there is not a scientific consensus on which is the most reliable animal model [41]. However, the rat seems to be a valid and reproducible model allowing the use of an adequate sample size to evaluate experimental conditions, compared to appropriate controls and sham groups to study the repair of non-spontaneous bisphosphonate osteonecrosis [42]. So, the present in vivo study has aimed to evaluate the risk-reduction strategy of MRONJ in different groups of rats exposed to different concentrations of BPs with different materials/antibiotic drug-delivery systems following a tooth extraction. Also, the clinical therapeutic dose used in patients was adjusted (calculated) for rats as previously described [42].

The study’s results confirmed that the protocol used was effective to induce osteonecrosis and that the association of DOX to HA reduced/impaired the necrosis at the moment of the analysis and allowed significantly higher bone volume density than HA without DOX. Moreover, the histological results of the entire socket showed no evidence of inflammatory infiltration in the HADOX group. The presentation of histological and histomorphometric results was divided into two stages. In the first stage, the groups without grafting were presented (clot with 4% ZA treatment and clot with 8% ZA treatment). Through this comparison, it was feasible to observe the effect of ZA on bone healing. Specifically, the 8% concentration showed a healing delay compared to the other two groups. Once the effect of ZA was proven, the grafted groups with and without ZA treatment were compared.

The histomorphometric analysis did not show that a significant reduction in the biomaterial was followed by new bone formation. Instead, more connective tissue occupied the dental socket in the HA group in the ZA 4 and 8%-treated animals. The decrease in bone formation associated with the low level of residual HA in the HA and HA ZA 4 and 8% samples can be attributed to two effects: the acute inflammatory process which lowers the local pH and accelerates the degradation of HA microspheres, and the exudate and suppuration which facilitates the removal of the microsphere fragments from the alveolus. The association of doxycycline with HA spheres modifies this clinical picture as it inhibits the inflammatory process and stabilizes the bone graft in the defect, increasing bone formation.

When comparing the results of the histomorphometric analysis between HADOX and clot groups (Sham groups), we observed that the amount of newly formed bone was predominantly associated with the clot groups (Sham groups). However, in the HADOX group, part of the socket was filled with biomaterial, which affected new bone formation. A potential advantage seen in the HADOX group is preserving the alveolar architecture, which is clinically very important when dental rehabilitation is required.

This study presents several limitations such as:Variations in drug-dosing schedules and regimens (potentially not relevant to human conditions);Variable effect of drug exposure or drug absorption and metabolization;Variable duration of follow-up, which may not correspond to disease manifestation in humans.

Despite the impact that these limitations might have, we should clearly acknowledge that the observed results support that HA appears to be a good carrier to deliver DOX locally in dental sockets after surgery, avoiding the development of necrosis and allowing higher levels of new bone compared to the HA-only groups.

Future research with a larger sample size to validate our findings is required. Before doing any clinical trials, large-animal models closely resembling human models in terms of anatomy of jaw, dentition, oral flora, bone structure, and remodeling properties might be needed to accurately determine the safety, efficacy, and effectiveness of the treatment compared with standard of care.

This could be the first step in the design of future clinical trials to evaluate this new potential alveolar dental surgery protocol.

In rats exposed to 4% and 8% of zoledronic acid, the histomorphometric analysis highlighted that NFB for the two experimental groups (HA and HADOX) was statistically different. In fact, rats treated with 4% ZA and HADOX exhibited significantly more NFB (28.38%) compared to those treated with 4% ZA and HA, where the NFB was significantly less (15.69%). This phenomenon was also observed in the equivalent groups exposed to 8% of ZA (*p* = 0.001). We also found a similar trend with regards to the percentage of connective tissue found in the experimental groups, where the connective tissue was much higher for both 4% and 8% HA groups compared to the HADOX. This favorable biological response reported in the HADOX experimental group is probably determined using antibiotics.

## 5. Conclusions

This study shows that the use of HADOX followed by a tooth extraction has a positive effect on alveolar bone repair in a dental extraction–rats model with high risk of MRONJ. The study’s results confirmed that the protocol used was effective to induce osteonecrosis and that the association of DOX to HA reduced/impaired the necrosis at the moment of the analysis and allowed significant higher bone volume density than HA without DOX.

HADOX showed improvement in the bone-repairing process, with a reduced connective tissue response. These characteristics could be attributed to the HADOX’s properties of being:-An anti-inflammatory;-An osteoclast inhibitor;-A fibroblast stimulator;-An anti-collagenolytic;-An antimicrobial agent.

In addition, this high biocompatibility was also clearly identified in subjects undergoing a high dose of bone-targeting drug therapy.

Furthermore, the study reports that in the HA groups the biomaterial is potentially lost due to exudate/suppuration, in comparison to the HADOX groups.

This promising result should be cautiously interpreted and may support the design of further clinical studies.

## Figures and Tables

**Figure 1 medicina-59-00046-f001:**
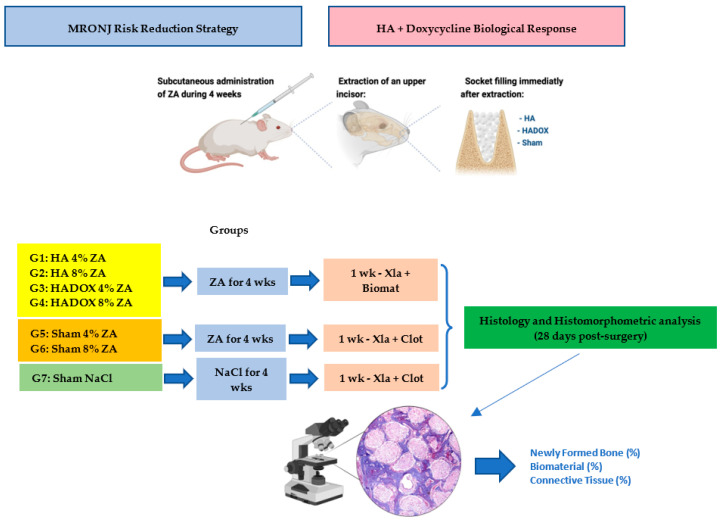
Experiment visual summary. Zoledronic acid (ZA); hydroxyapatite (HA); hydroxyapatite-containing doxycycline (HADOX); weeks (wks); week (wk); dental extraction (xla).

**Figure 2 medicina-59-00046-f002:**
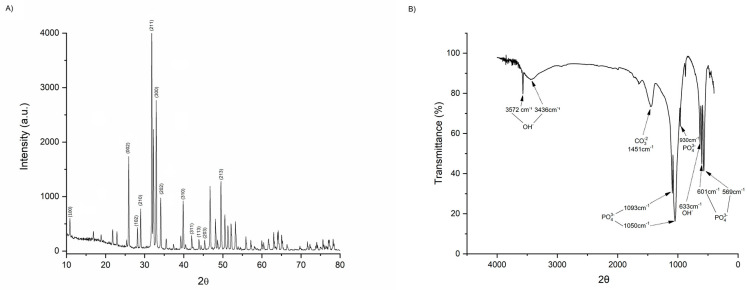
XRD structure of HADOX microsphere presenting the expected features of a stoichiometric hydroxyapatite configuration (**A**) and HADOX microsphere FTIR fields exhibiting (PO4)3− and OH− bands (**B**).

**Figure 3 medicina-59-00046-f003:**
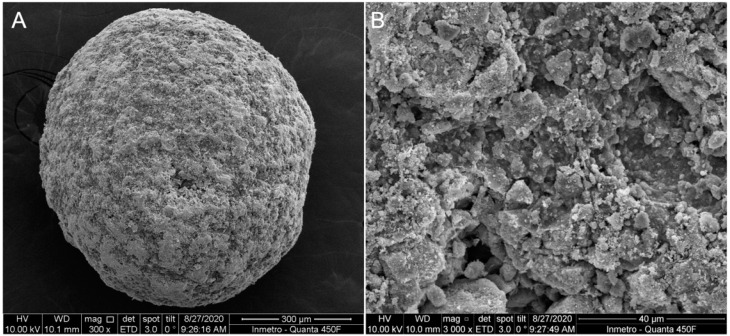
SEM images of HA microspheres at lower magnification (**A**) and higher magnification (**B**).

**Figure 4 medicina-59-00046-f004:**
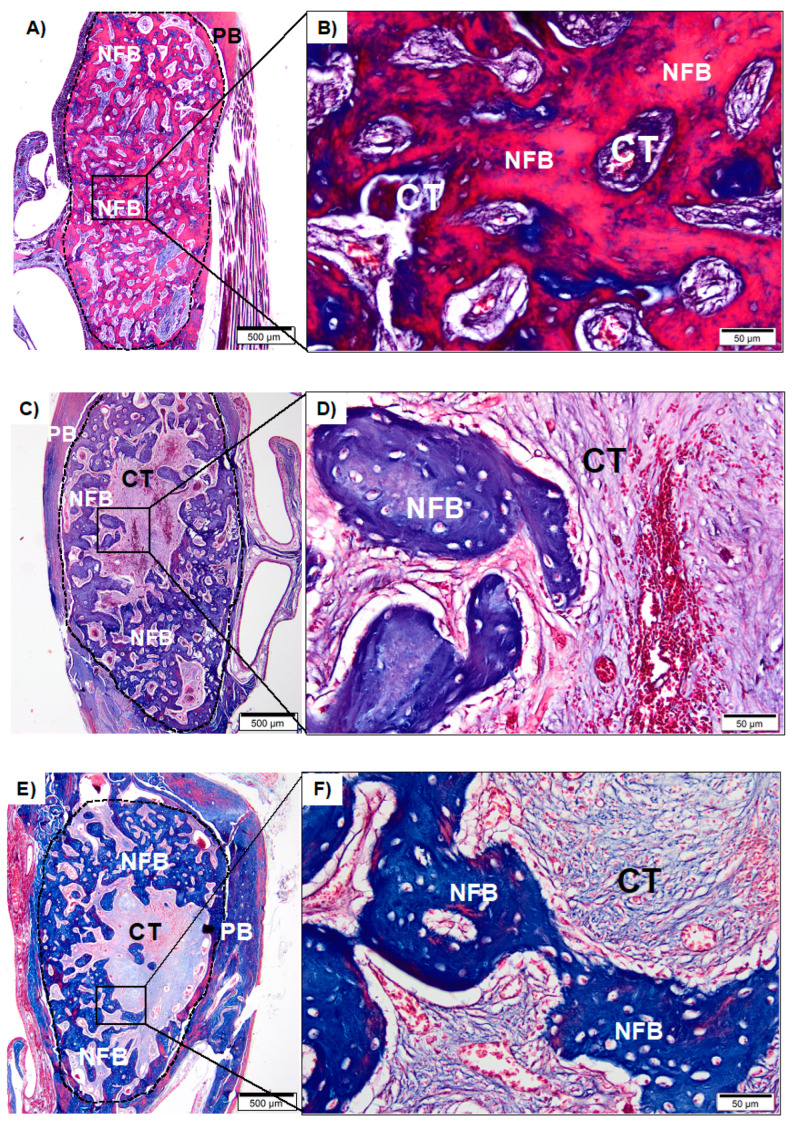
(**A**): Photomicrograph of Sham group (without ZA treatment) after 28 days at 20× magnification. (**B**): Sham group in detail at 40× magnification. (**C**): Photomicrograph of Sham 4% ZA after 28 days at 20× magnification. (**D**): Sham 4% ZA in detail at 40× magnification. (**E**): Photomicrographs of Sham 8% ZA after 28 days at 20× magnification. (**F**): Sham 8% ZA in detail at 40× magnification. Newly formed bone (NFB); connective tissue (CT) pre-existing bone (PB). (**A**,**C**,**E**): scale bar 500 µm; (**B**,**D**,**F**): scale bar: 50 µm. Staining with Masson’s Trichrome.

**Figure 5 medicina-59-00046-f005:**
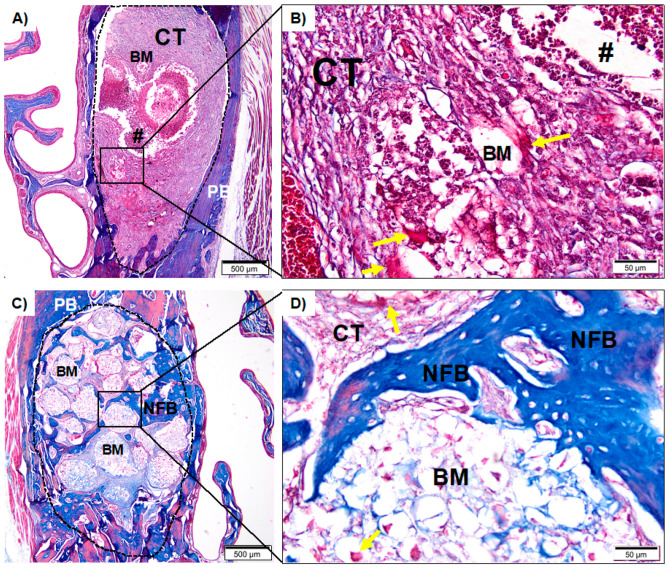
(**A**): Photomicrograph of HA 4% ZA group after 28 days at 20× magnification. (**B**): HA 4% ZA group in detail at 40× magnification. (**C**): Photomicrograph of HADOX 4% ZA group after 28 days at 20× magnification. (**D**): HADOX 4% ZA in detail at 40× magnification. Newly formed bone (NFB); connective tissue (CT) pre-existing bone (PB); biomaterial (BM); multinucleated giant cells (yellow arrows). (**A**,**C**): scale bar 500 µm; (**B**,**D**): scale bar: 50 µm. Staining with Masson’s Trichrome.

**Figure 6 medicina-59-00046-f006:**
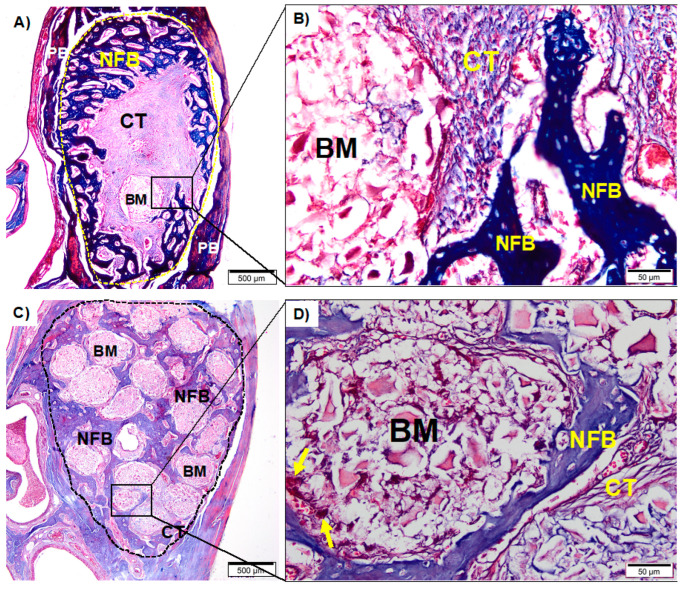
(**A**): Photomicrograph of HA 8% ZA group after 28 days at 20× magnification. (**B**): HA 8% ZA group in detail at 40× magnification. (**C**): Photomicrograph of HADOX 8% ZA group after 28 days at 20× magnification. (**D**): HADOX 8% ZA in detail at 40× magnification. Newly formed bone (NFB); connective tissue (CT) pre-existing bone (PB); biomaterial (BM); multinucleated giant cells (yellow arrows). (**A**,**C**): scale bar 500 µm; (**B**,**D**): scale bar: 50 µm. The interest area is circulated by the dotted staining with Masson’s Trichrome.

**Figure 7 medicina-59-00046-f007:**
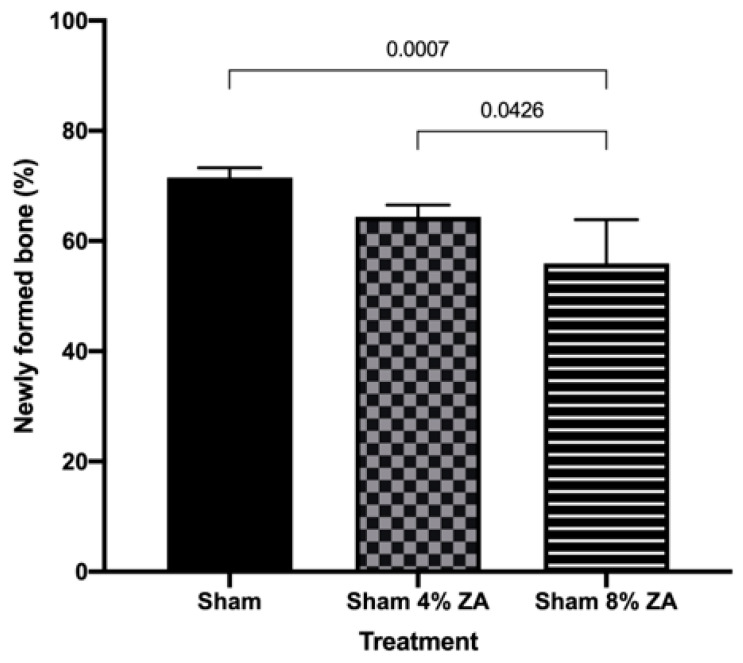
Newly formed bone in the Sham group (without ZA treatment), Sham 4%ZA and Sham 8% ZA treatments. The values are presented as mean ± C.I. The statistical differences between treatments are represented by horizontal bars. There were no statistical differences between clot 4% ZA and clot 8% ZA (ANOVA and Tukey’s post-test, *p* <0.05).

**Figure 8 medicina-59-00046-f008:**
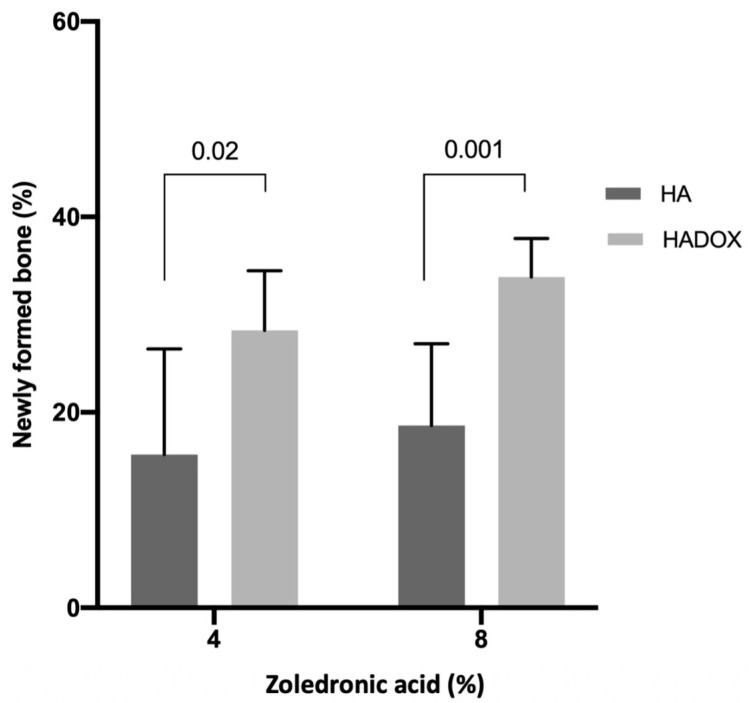
Newly formed bone in HA and HADOX groups treated with 4% and 8% ZA. The values are presented as mean ± C.I. The statistical differences between the groups in the same ZA treatment are represented by horizontal bars. There were no statistical differences (*p* > 0.05) between 4% and 8% ZA treatment in the HA and HADOX groups (no difference is represented by the same letter) (Student’s *t*-test, *p* > 0.05).

**Figure 9 medicina-59-00046-f009:**
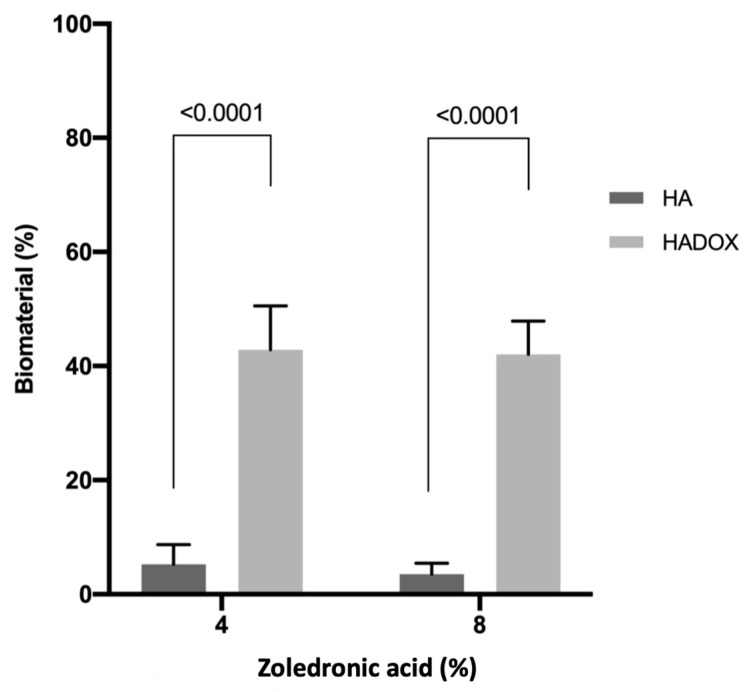
Biomaterial volume density in the HA and HADOX groups treated with 4% and 8% ZA. The values are presented as mean ± C.I. The statistical differences between the groups in the same ZA treatment are represented by horizontal bars. There were no statistical differences (*p* > 0.05) between 4% and 8% ZA treatment in the HA and HADOX groups (no difference is represented by the same letter) (Student’s *t*-test, *p* > 0.05).

**Figure 10 medicina-59-00046-f010:**
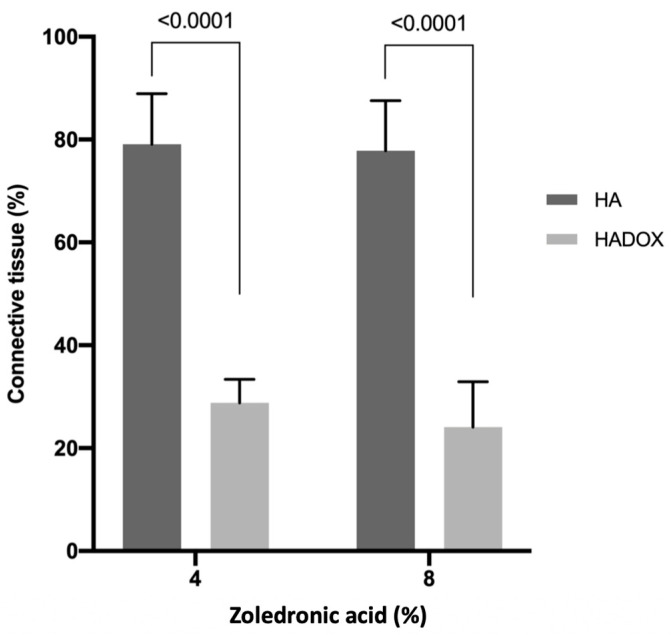
Connective tissue volume density in HA and HADOX groups with 4% and 8% ZA treatment. The values are presented as mean ± C.I. The statistical differences between the groups in the same ZA treatment are represented by horizontal bars. There were no statistical differences (*p* > 0.05) between 4% and 8% ZA treatment in the HA and HADOX groups (no difference is represented by the same letter) (Student’s *t*-test, *p* > 0.05).

**Table 1 medicina-59-00046-t001:** Summary of the result findings. Mean%, confidence interval (CI) and *p* values of newly formed bone, remaining biomaterial and connective tissue parameters in Sham (clot), HA and HADOX groups. Statistical difference comparing clot groups with the same ZA (%). Statistical difference comparing HA groups with the same ZA (%).

	Newly Formed Bone			Biomaterial			Connective Tissue		
	Mean (%)	CI (95%)	*p*-Value	Mean (%)	CI (95%)	*p*-Value	Mean (%)	CI (95%)	*p*-Value
**Clot**									
**ZA 4%**	64.40	61.75-67.06					35.60	32.94–38.25	
**ZA 8%**	55.94	46.10–65.78					44.06	34.22–53.90	
**HA**									
**ZA 4%**	15.69	4.89–26.48	a (<0.0001)	5.26	1.80–8.71		79.06	69.20–88.91	a (<0.0001)
**ZA 8%**	18.65	10.29–27.02	a (<0.0001)	3.5	1.54–5.45		77.84	68.11–87.58	a (<0.0001)
**HADOX**									
**ZA 4%**	28.38		a (<0.0001); b (0.008)		35.12–50.54	b (<0.0001)	28.79	24.20–33.38	b (<0.0001)
**ZA 8%**	33.85		a (<0.0001); b (0.002)		36.23–47.88	b (<0.0001)	24.10	15.30–32.89	a (0.0002), b (<0.0001)

## Data Availability

Data are included within the article.

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
