# Peer review of "The Use of Hydroxyapatite Loaded with Doxycycline (HADOX) in Dentoalveolar Surgery as a Risk-Reduction Therapeutic Protocol in Subjects Treated with Different Bisphosphonate Dosages"

_medicina, 2022, doi:10.3390/medicina59010046_

Round 1
Reviewer 1 Report
The study seems very interesting and genuine, however the authors should address the following points to improve the overall quality of the manuscript:
- The abstract should be non-structured with specific word limit (please review the authors guidelines).
- Please add the research hypothesis/hypotheses at the end of the introduction section.
- Please indicate the number of rats used for the pilot study.
- The authors should clearly mention the limitations of this study and suggest several directions for future research.
- Line 472 belongs to conclusion section.
- The conclusion should be expanded to reflect the potential outcomes of this study and can be summarized in bullets for clarity.
Author Response
Reviewer 1
The study seems very interesting and genuine, however the authors should address the following points to improve the overall quality of the manuscript:
- The abstract should be non-structured with specific word limit (please review the authors guidelines).
Answer: Thanks for the remark. We have adjusted the abstract structure accordingly.
- Please add the research hypothesis/hypotheses at the end of the introduction section.
Answer: Thanks for the remark. We have included the hypothesis in the introduction part.
- Please indicate the number of rats used for the pilot study.
Answer: Thanks for the remark. We have included a paragraph on the pilot study in material and methods part.
- The authors should clearly mention the limitations of this study and suggest several directions for future research.
Answer: Thanks for the remark. We have included two paragraphs that suggested the study limitations and we suggested a direction for future studies.
- Line 472 belongs to conclusion section.
Answer: Thanks for this remark. Line 472 has been now moved to the conclusion part.
- The conclusion should be expanded to reflect the potential outcomes of this study and can be summarized in bullets for clarity.
Answer: Thanks for the remark. We have included a paragraph with some bullet points that explain the associated outcomes of the study.

Reviewer 2 Report
Respectful authors,
Thank you for submitting this article and congratulations on your work.
Still, I have some concerns.
1. Please, explain better how doxycycline (DOX) was incorporated into the hydroxyapatite (HA) microspheres.
2. What was the releasing rate of DOX from the microspheres during the 4 weeks period? If the release takes place rapidly, are there any concerns about reaching toxic levels? Maybe an experiment that evaluates DOX release would be nice to be performed (as DOX release in PBS and HPLC analysis of the extraction solutions. See Miron, A. E., Moldovan, M., Prejmerean, C. A., Prodan, D., Vlassa, M., Filip, M., ... & Moldovan, M. A. (2020). New Antimicrobial Biomaterials for the Reconstruction of Craniofacial Bone Defects. Coatings, 10(7), 678. https://www.mdpi.com/2079-6412/10/7/678)
3. It is certain that HADOX improves the outcome after extraction as compared to HA alone, but I do not clearly picture the advantages of using HADOX instead the clot itself. Could you better explain the benefits?
Cordially,
Author Response
Reviewer 2
Respectful authors,
Thank you for submitting this article and congratulations on your work.
Still, I have some concerns.
- Please, explain better how doxycycline (DOX) was incorporated into the hydroxyapatite (HA) microspheres.
Answer: Thanks for the remark. A paragraph has been added to the bone substitute synthesis and characterization.
- What was the releasing rate of DOX from the microspheres during the 4 weeks period? If the release takes place rapidly, are there any concerns about reaching toxic levels? Maybe an experiment that evaluates DOX release would be nice to be performed (as DOX release in PBS and HPLC analysis of the extraction solutions. See Miron, A. E., Moldovan, M., Prejmerean, C. A., Prodan, D., Vlassa, M., Filip, M., ... & Moldovan, M. A. (2020). New Antimicrobial Biomaterials for the Reconstruction of Craniofacial Bone Defects. Coatings, 10(7), 678. https://www.mdpi.com/2079-6412/10/7/678)
Answer: Thanks for the comment. This represents one of the limitations of the study. However, we performed a pilot study with no adverse effects observed. In addition, our research group recently published a paper on this regard and it was found that DOX uptake was 28.2 ± 4.5 mgDOX/mgHA for HADOX0 microspheres samples. In the study, it was observed an initial drug release of 49.15 μg/mL (~20%) in the first 24 hr, followed by a continuous drug release, with a 60% of DOX remaining retention into HA microspheres after 9 days. (Soriano-Souza, C.; Valiense, H.; Mavropoulos, E.; Martinez-Zelaya, V.; Costa, A.M.; Alves, A.T.; Longuinho, M.; Resende, R.; Mourão, C.; Granjeiro, J.; Rocha-Leao, M.H.; Rossi, A.; Calasans-Maia, M. Doxycycline containing hydroxyapatite ceramic microspheres as a bone-targeting drug delivery system. J Biomed Mater Res B. 2020, 108, 1351-1362.)
- It is certain that HADOX improves the outcome after extraction as compared to HA alone, but I do not clearly picture the advantages of using HADOX instead the clot itself. Could you better explain the benefits?
When comparing the results of the histomorphometrical analysis between HADOX and clot groups, we observed that the amount of newly formed bone was predominantly associated with the clot groups. However, in the HADOX group, part of the socket was filled with biomaterial, which affected new bone formation. A potential advantage seen in the HADOX group is preserving the alveolar architecture, which is clinically very important in case dental rehabilitation is required. The authors included this information in the discussion part.
